# Enzymatic Synthesis of Ascorbic Acid-Ketone Body Hybrids

Valentina Venturi [1], Lindomar Alberto Lerin [2], Francesco Presini [2], Pier Paolo Giovannini [2,*], Martina Catani [2], Alessandro Buratti [2], Nicola Marchetti [2], Latha Nagamani Dilliraj [2] and Simona Aprile [2]

[1] Department of Environmental and Prevention Sciences, University of Ferrara, Via Luigi Borsari, 46, 44121 Ferrara, Italy; valentina.venturi@unife.it
[2] Department of Chemistry, Pharmaceutical and Agricultural Sciences, University of Ferrara, Via Luigi Borsari, 46, 44121 Ferrara, Italy; lindomaralberto.lerin@unife.it (L.A.L.); francesco.presini@unife.it (F.P.); martina.catani@unife.it (M.C.); alessandro.buratti@unife.it (A.B.); nicola.marchetti@unife.it (N.M.); latha7481@yahoo.com (L.N.D.); simona.aprile@unife.it (S.A.)
* Correspondence: pierpaolo.giovannini@unife.it; Tel.: +39-053-297-4532

**Abstract:** Molecular hybrids obtained by connecting two or more bioactive molecules through a metabolizable linker are used as multi-target drugs for the therapy of multifactorial diseases. Ascorbic acid, as well as the ketone bodies acetoacetate and (*R*)-3-hydroxybutyrate, are bioactive molecules that have common fields of application in the treatment and prevention of neurodegenerative diseases and cardiac injuries as well. In spite of this, the preparation of ascorbic acid ketone body hybrids is uncovered by the literature. Herein, we report the lipase-catalyzed condensation of methyl acetoacetate with ascorbic acid, which affords the 6-*O*-acetoacetyl ascorbic acid in quantitative yield. The same approach, employing the methyl (*R*)-3-hydroxybutyrate in place of the methyl acetoacetate, allows the preparation of the 6-*O*-(*R*)-3-hydroxybutyryl ascorbic acid in 57% yield. A better result (90% overall yield) is achieved through the lipase-catalyzed coupling of ascorbic acid with methyl (*R*)-3-*O*-methoxymethyl-3-hydroxybutyrate followed by the cleavage of the MOM protecting group. The two novel products are fully characterized and additional information on the antioxidant activity of the new products is also given.

**Keywords:** molecular hybrids; ascorbyl esters; lipase; biocatalysis; antioxidant activity





## 1. Introduction

Humans have exploited for millennia the properties of plant-derived foods and beverages to preserve their health, and the knowledge growth in various scientific fields has led to the identification of the substances responsible for the protective or curative effects. More recently, thanks to in vitro as well as in vivo experiments, the mechanisms by which many plant-derived bioactive compounds produce health benefits have been discovered [1,2]. It is worth noting that a number of bioactive compounds are employed in the treatment or prevention of multifactorial diseases, namely illnesses with complex etiopathology due to the involvement of multiple organ systems and tissues [3]. The treatment of such diseases moved from the traditional single-drug therapy toward multidrug approaches that include the administration of drug cocktails or multicomponent drugs obtained by the co-formulation of more than one active ingredient [4]. An alternative emerging strategy is the design of multi-target drugs that integrate multiple pharmacophores into a single drug molecule in order to simultaneously act at multiple sites of relevance to a disease [5,6]. Multi-target drugs can be prepared by connecting two or more drugs through a stable or metabolizable linker or by merging the haptophoric moieties of different drugs. The two strategies afford hybrid and chimeric drugs, respectively [3], although this specification is not strictly followed in the literature and the term molecular hybrid is often used to indicate multi-target drugs achieved through both connection strategies [3,7]. Many bioactive natural products have been employed as a starting scaffold in multi-target drug discovery

studies [8,9], and among these, ascorbic acid plays a central role by virtue of its well-known antioxidant and free-radical scavenging activity beneficial for the therapy or prevention of various oxidative-stress-related chronic diseases such as cancer, diabetes, atherosclerosis, cardiovascular diseases, inflammation, and neurodegenerative pathologies [10]. In addition to being a general antioxidant, recent advances have identified the role of ascorbic acid in the epigenetic regulation of gene activity as well as the strong and selective antitumor and antiviral activity of several ascorbic acid derivatives [10]. By virtue of its accessibility and reactivity as well as of its lower involvement in the mechanism responsible for the antioxidant activity [11], the hydroxyl group in position 6 represents a privileged site for the anchoring of pharmacophores or bioactive moieties to ascorbic acid. In fact, a number of 6-*O*-acyl derivatives have been synthetized with different purposes. For instance, ascorbic phenolic acid hybrids with increased antioxidant activity and weaker antiaggregant action with respect to ascorbic acid have been obtained through acid-catalyzed esterification of the 6-hydroxyl group of ascorbic acid with protocatechuic, gallic, and caffeic acid [12]. Furthermore, several 6-*O*-ascorbyl esters of long-chain fatty acids employed as antioxidants and surfactants in cosmetic, nutraceutical, and food products [13] have shown potent inhibitory activity on enzymes involved in the onset of various diseases. For instance, 6-*O*-ascorbyl palmitate inhibits lipoxygenases responsible for the biosynthesis of leukotriene and ROS implicated in the pathophysiology of inflammatory disorders [14], while the 6-*O*-ascorbyl hexadecanoate was 1.500 times more active than ascorbic acid in inhibiting the *Streptococcus pneumoniae* hyaluronidase [15]. In these cases, the alkyl chain increases the affinity of the inhibitor for the enzyme. On the other hand, 6-*O*-ascorbyl esters of medium-chain fatty acids instead (C6-C11) showed better anti-tumor-promoting effects on the activation of the Epstein–Barr virus early antigen [16]. A limited number of studies report on the biological activities of short-chain carboxylic acids 6-*O*-ascorbyl esters. A slightly higher antioxidant activity with respect to ascorbic acid has been reported for the acetyl, 4-chlorobutanoyl and 5-chloropentaoyl derivatives [17–20]. Among the bioactive short-chain carboxylic acids, the ketone bodies acetoacetate and (*R*)-3-hydroxybutyrate are receiving increasing attention [21]. They are produced in the liver through the beta-oxidation of free fatty acids during hypoglycemic conditions and have demonstrated that the increased ketone bodies' blood concentration induced by fasting, diabetes, or strenuous physical exercise plays a beneficial role in the control of oxidative stress and inflammation associated with several chronic neurological disorders and cardiovascular diseases. As a consequence, today, the administration of exogenous ketone bodies is not only considered a strategy for increasing physical and cognitive performance [22], but also for the treatment and prevention of neurodegenerative diseases [23] and cardiac injuries [24]. In spite of this, until now, the preparation of ascorbic acid ketone body hybrids is practically unexplored [25]. In order to close this gap, making available to the scientific community new potentially interesting hybrid compounds, we herein discuss the results obtained in the enzymatic synthesis of the 6-*O*-ascorbyl esters of acetoacetic and (*R*)-3-hydroxybutyric acids. Thanks to lipase efficiency and selectivity, enzymatic and chemo-enzymatic synthetic routes were designed and implemented for the preparation of both target compounds. The new ketone bodies' ascorbic acid hybrids were fully characterized and preliminary tests on their antioxidant activity were performed as well.

## 2. Results and Discussion

Most of the above-cited 6-*O*-acyl derivatives have been obtained by the acid- or base-catalyzed condensation of ascorbic acid with free carboxylic acids or acyl chlorides, respectively. As an alternative, alkyl esters have been employed as the acylating agents in reactions promoted by base or enzyme catalysts (Table 1). We identified the lipase B from *Candida antarctica* immobilized on macroporous acrylic resin Novozym 435® (N 435) as the suited biocatalyst for the transesterification of ketone body methyl esters with ascorbic acid as both substrates have already been reported as accepted by this enzyme. In addition, the

N 435 is probably the most easily purchasable biocatalyst worldwide and many studies demonstrate its reusability [26].

**Table 1.** Acyl donors and catalysts employed in the syntheses of 6-*O*-ascorbyl esters.

| Acyl Donors | Catalyst | Reference |
|---|---|---|
| Phenolic acids | $H_2SO_4$ | [12] |
| LCFA [a] ($C_{12}$-$C_{18}$) | $H_2SO_4$ | [13] |
| SCFA [b], MCFA [c] and LCFA ($C_2$-$C_{18}$) | Lipase [d] | [18] |
| SCFA, MCFA, LCFA ($C_2$-$C_{18}$) chlorides or methyl esters [e] | $C_5H_5N$ [f]; lipase [g] | [15] |
| vinyl acetate | Lipase [h] | [20] |
| SCFA, MCFA and LCFA ($C_4$-$C_{17}$) [i] methyl esters | $H_2SO_4$ | [16] |
| MCFA and LCFA ($C_6$-$C_{18}$) | $H_2SO_4$ | [19] |
| 4-Cl-butanoyl and 5-Cl-pentanoyl chlorides | $C_5H_{11}N$ | [19] |

[a] LCFA = Long-Chain Fatty Acid; [b] SCFA = Short-Chain Fatty Acid; [c] MCFA = Medium-Chain Fatty Acid; [d] form *Staphylococcus xylosus*; [e] some of the acyl esters bear a terminal hydroxyl group protected as phenyl, benzyl, biphenyl, or *p*-phenyl benzyl ethers; [f] pyridine was used with the acyl chloride donors; [g] lipase C from *Candida antarctica* was used with the methyl ester donors; [h] from *Thermomyces lanuginosus*; [i] also with $C_7$ and $C_9$ bearing a methyl substituent in position 2.

The first acyl donor employed was the methyl acetoacetate (compound **1**, Scheme 1). In order to obtain a homogeneous solution of ascorbic acid (**2**) and **1** (8 equiv.), a polar solvent was needed and *tert*-butanol was chosen by virtue of the known inactivity of the N 435 in transesterification of tertiary alcohols [27]. Furthermore, 5 Å molecular sieves were added to the reaction mixture for the trapping of the methanol coproduced by transesterification. After the addition of the biocatalyst (80% *w/w* with respect to ascorbic acid), the reaction mixture was gently shaken at 60 °C and the reaction course was monitored by $^1$H-NMR. The conversion was determined from the ratio between the doublets centered at 3.40 and 4.09 ppm attributable to the C6 methylene group of the free and esterified ascorbic acid, respectively. A maximum conversion of 60% was reached after 48 h. A parallel blank experiment performed in the absence of the biocatalyst showed the absence of any reaction product at the same time. The reaction work-up was conducted by removing the enzyme and the molecular sieves by filtration and evaporating under reduced pressure the solvent and the excess of **1** as well. The residue was chromatographed on silica gel using chloroform/methanol 7:1 as the eluent.

**Scheme 1.** N-435-catalyzed synthesis of 6-*O*-acetoacetyl ascorbic acid **3**: (**a**) reaction volume 5 mL, **1** (0.91 M), **2** (114 mM), *t*-butanol (4 mL), 5 Å molecular sieves (0.8 g), N 435 (40 mg), yield determined by $^1$H-NMR (less than 10% after column chromatography); (**b**) reaction volume 5 mL, **1** (1.82 M), **2** (114 mM), *t*-butanol (4 mL), 5 Å molecular sieves (0.8 g), N 435 (40 mg), yields calculated on the amount of product **3** isolated after evaporation of the solvent and of the unreacted **1**.

Unfortunately, the pure product **3** was recovered in less than 10% of isolated yield. In an attempt to reach the complete conversion of **2** avoiding chromatographic separation of the product **3** from unreacted **2**, the **1** to **2** molar ratio was increased from 8:1 to 16:1.

Under this condition, after 6 h, the complete conversion of **2** into the expected product was achieved and the pure compound **3** was obtained in 95% yield by atmospheric pressure distillation of the solvent followed by vacuum distillation to recover the unreacted methyl ester **1**.

Supported by this result, we moved to study the enzymatic transesterification of the methyl (*R*)-3-hydroxybutyrate (compound **4**, Figure 1) with ascorbic acid. The first attempt was conducted under the same conditions adopted for the synthesis of the product **3,** namely 16 equivalents of the acyl donor. In this case also, a parallel blank experiment performed in the absence of the biocatalyst showed the absence of any reaction product, while the $^1$H NMR analyses of the biocatalyzed reaction mixture showed the presence of an unexpected set of signals in addition to those attributable to the desired product **5** and to the unreacted ascorbic acid.

**Figure 1.** N-435-catalyzed synthesis of 6-*O*-(R)-3-hydroxybutyryl ascorbic acid **5**: with one, two, and three equivalents of the acylating agent **4** (conditions A, B, and C, respectively). Reaction conditions: **2** (0.57 mmol), **4** (0.57 or 1.14 or 171 mmol), *t*-butanol (8 mL), 5 Å molecular sieves (0.8 g), N 435 (80 mg). The composition of the crude reaction mixtures was determined by $^1$H-NMR analysis. The values are the means of three independent experiments.

In particular, a multiplet centered at 5.09 ppm suggested the presence of 3-hydroxybutyrate oligomers [28]. The results of the ESI-MS analysis (Supplementary Materials Figure S4) of the reaction mixture confirmed the presence of a higher homolog of **5** where the ascorbyl moiety is esterified with a dimer of **4** (compound **6**, Figure 1). The effect of the substrates' molar ratio on the conversion, yield, and selectivity was studied by performing the reaction in the presence of 1, 2, or 3 equivalents of **4**. The composition of the crude reaction mixture was monitored by $^1$H NMR by comparing the integrals of the signals attributable to the H-4 proton of the species **6** (4.65 ppm), **5** (4.67 ppm), and **2** (4.68 ppm). Figure 1 shows the composition of the reaction mixtures corresponding to the higher yields reached by the three reactions (Supplementary Materials Figure S5). It can be observed that using 2 equivalents of **4** (condition B, Figure 1), the maximum yield of 60% was reached (after 3.5 h) without the formation of the byproduct **6** (Supplementary Materials Figure S5). On the other hand, the best yields reached by the reactions performed with 1 and 3 equivalents of **4** (conditions A and C, Figure 1) were 50% and 60%, respectively (obtained after about 4 and 3 h); however, in these cases, the mixtures contained 11% and 22% of the byproduct **6,**

respectively (Supplementary Materials Figure S5). Figure 2 shows the time course of the reaction performed with 2 equivalents of 4. As can be observed, the byproduct **6** started yielding after about 4 h and its concentration linearly increased at longer reaction times, making crucial the monitoring of the reaction course (Supplementary Materials Figure S6). In the attempt to reduce the formation of **6,** the reaction was also performed at the lower temperature of 50 °C but, in this case, only a kinetic delay was observed with an equivalent slowing down of both the rates of formation of **5** and **6** (data not shown).

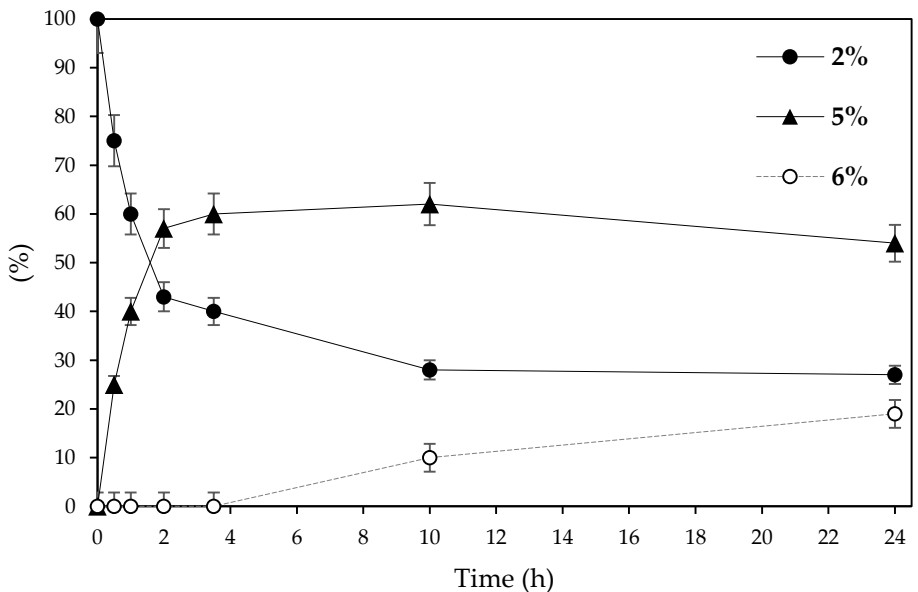

**Figure 2.** Time course of the N-435-catalyzed synthesis of 6-*O*-(R)-3-hydroxybutyryl ascorbic acid **5** performed with **4**. The values are the means of three independent experiments.

As for the product **3**, the purification on the silica gel column of the compound **5** afforded a very low chromatographic yield. Therefore, the pure compound **5** was isolated from the unreacted ascorbic acid through reversed-phase preparative liquid chromatography by using a $C_{18}$ stationary phase and a mixture of water and an organic modifier + 0.05%$_{v/v}$ formic acid (FA) as the mobile phase. Both acetonitrile (ACN) and ethanol (EtOH) were tested as organic modifiers, finding comparable results in terms of purity and yield of the process. However, the use of EtOH could be preferred over ACN as it allows for improving the sustainability of the process and to match the goals of green chemistry. Despite this appreciable result, we decided to investigate an alternative synthetic strategy that could afford the complete conversion of the ascorbic acid without the need of a careful monitoring of the reaction course. We thought to pursue this goal through the employment of *O*-protected analogs of **4**. In this view, we started using the t-butyl-dimethyl silyl (TBDMS) derivative **7a** (Scheme 2) obtained as previously described [29]. The reaction performed with **7a** under the same reaction condition adopted with the unprotected acyl donor **4** did not afford any product. This was probably due to the hindrance of the TBDMS group that prevented the access of the substrate into the N 435 active site. Therefore, a second attempt was conducted by using the less bulky trimethyl silyl (TMS) derivative **7b** (Scheme 2). In this case, however, the reaction afforded a mixture of the product **5** together with the byproduct **6,** because of the lability of the TMS group under the employed reaction conditions. Searching for an alternative protecting group able to afford a less hindered and more stable acylating reagent, we prepared the 3-*O*-methoxymethyl (MOM) derivative **7c** by following a known procedure [30]. The MOM group was chosen also because it can be cleaved under acidic conditions compatible with the ester group [31]. The transesterification of **7c** (5 equivalents) with ascorbic acid performed in the presence of N 435 showed that after 24 h, the complete conversion of **2** into the desired derivative **8c** was obtained in a crude form by simply removing the solvent and the excess of **7c** by vacuum distillation.

Finally, the desired product **5** was obtained by treating a methanol/water (5:1) solution of the crude **8c** with Amberlite IR120 at 50 °C for 6 h. This second synthetic strategy allowed for pushing the ascorbic acid conversion to completion, avoiding both the formation of byproducts and the need of chromatographic purification of the final product **5,** even if the increased number of steps and the lower overall atom economy need to be taken into account in comparing the two proposed synthetic strategies.

**7a-c**
**7a** R = TBDMS          **8a** (n.d.)          —
**7b** R = TMS            **8b** (n.d.)          —
**7c** R = MOM            **8c** (95%)           **5** (90%)

**Scheme 2.** Chemo-enzymatic synthesis of **5** through the *O*-protected intermediate **7a–c**. Reaction conditions: **7a–c** (2.85 mmol, 5 equiv.), **2** (0.57 mmol), *t*-butanol (8 mL), 5 Å molecular sieves (0.8 g), and N 435 (80 mg); n.d. = not detected.

To completely assess the synthetic strategies, a study on the reusability of the biocatalyst was conducted. The optimized synthesis of the acetoacetyl derivative **3** (Scheme 1) was used as the benchmark reaction. In this case, when the reaction was finished, the molecular sieves were removed from the reaction mixture by filtration on a 7-mesh sifter and then the biocatalyst was recovered by filtering the resulting suspension on a filter paper. The recovered N 435 was washed twice with acetone, dried under reduced pressure (20 mm Hg, 40 °C, 1 h), and added to a fresh reaction mixture. The biocatalyst was reused for ten reaction cycles, showing a slight decrease in efficiency after each cycle that led to a 15% loss of activity after the ninth cycle.

The ability of the new hybrids to act as radical scavengers was assessed in order to furnish the preliminary information about their functional features although the potential of the new compounds as multitarget drugs is not strictly related to their antioxidant power. The antioxidant activities of the compounds were determined by using a Photochemiluminescence (PCL) method [32], which uses an optical excitation (365 nm) of a photosensitive compound with UV-light, thus activating a chemical reaction resulting in the production of photo-energy to be measured, and a spectrophotometric Trolox Equivalent Antioxidant Capacity (TEAC) assay by means of a DPPH assay [33]. As reported in Table 2, both PCL and DPPH assays evidenced the same trend of measured TEAC: 2 > 5 > 3. The methyl acetoacetate **1** and the methyl (*R*)-3-hydroxybutyrate **4** did not show significant antioxidant activities, as evidenced by extremely low values.

Although DPPH overestimated the PCL results, they both led to similar conclusions. Product **3** retained between 10% (PCL) and 12.4% (DPPH) of the antioxidant capacity exhibited by ascorbic acid. Product **5**, instead, was able to retain about a 2.5 times larger antioxidant capacity (i.e., 24.8% by PCL and 31.5% by DPPH).

**Table 2.** Antioxidant activities of the products **3** and **5** and of the reagents **1, 2,** and **4**.

| Compound | PCL ($nmol_{TE}/mg_{ds}$) [1] | DPPH ($nmol_{TE}/mg_{ds}$) [1] |
|---|---|---|
| 1 | $(4.4 \pm 0.1)\ 10^{-4}$ | $(10 \pm 1)\ 10^{-3}$ |
| 2 | $7.5 \pm 0.3$ | $15.5 \pm 0.6$ |
| 3 | $0.75 \pm 0.02$ | $1.92 \pm 0.02$ |
| 4 | $(9.8 \pm 0.1)\ 10^{-5}$ | $(3.7 \pm 1.2)\ 10^{-3}$ |
| 5 | $1.86 \pm 0.05$ | $4.88 \pm 0.09$ |

[1] $nmol_{TE}/mg_{ds}$: nmoles of trolox equivalent per mg of dry sample.

## 3. Materials and Methods

### 3.1. General Information

All commercially available reagents were used without further purification, unless otherwise stated. The CAL-B Novozym® 435 (N 435) was a Novozymes product. TLC analyses were performed on silica gel 60 F254 with detection by charring with phosphomolybdic acid. Silica gel 60 (230–400 mesh) was employed for flash column chromatography. Compounds **7a** and **7c** were prepared as described in references [27,28], respectively. $^1$H and $^{13}$C NMR spectra were recorded on 300 and 400 MHz spectrometers at room temperature using the solvents specified below. Chemical shifts (δ) are reported in ppm relative to residual solvent signals. Optical rotations were measured at $20 \pm 2$ °C in the solvents specified below and the $[\alpha]^{20}_D$ values are given in $10^{-1}$ deg cm$^2$ g$^{-1}$. High-resolution mass spectra (HRMS) were recorded in positive ion mode with an Agilent 6520 HPLC-Chip Q/TOF-MS nanospray system using a time-of-flight, quadrupole or hexapole unit to produce spectra. The reversed-phase preparative liquid chromatography was performed on an ÅKTA pure 25 L instrument (GE Healthcare, Uppsala, Sweden) equipped with a fraction collector and a UV detector and operated through the Unicorn software. The column employed was a 250 × 4.6 mm (L × I.D.) Daisogel SP-120-10-ODS-BIO (DAISO Fine Chem GmbH, Dusseldorf, Germany) packed with 10 μm C18 particles with a 120 Å pore size. The total column volume (CV) calculated as the geometrical volume of the column was 4.2 mL. The detector wavelength was set at 254 nm.

### 3.2. Synthesis of 6-O-acetoacetyl Acorbic Acid 3

The ascorbic acid **2** (100 mg, 0.57 mmol) was added to *t*-butanol (8 mL) and the mixture was shaken (orbital shaker) at 60 °C until **2** was completely dissolved. Then, methyl acetoacetate **1** (1.06 g, 9.12 mmol), 5 Å molecular sieves (0.8 g), and N 435 (80 mg) were added to the solution and the mixture was shaken at 60 °C for 6 h. The mixture was passed through a 7-mesh filter in order to remove the molecular sieves and then through filter paper to recover the N 435 (both filters were washed with chloroform). After solvent evaporation under reduced pressure (20 mmHg), the excess of **1** was recovered by vacuum distillation (2 mm Hg), leaving the pure product **3** (140 mg, 0.54 mmol) in 95% yield. $[\alpha]^{20}_D$ = +8.2 (c 0.7, CH$_3$OH). $^1$H NMR (400 MHz, dmso) δ 4.67 (d, *J* = 1.8 Hz, 1H), 4.09 (d, *J* = 6.7 Hz, 2H), 3.95 (td, *J* = 6.7, 1.8 Hz, 1H), 3.62 (s, 2H), 2.17 (s, 3H); 13C NMR (101 MHz, dmso) δ 202.00, 170.80, 167.51, 152.79, 118.56, 75.33, 65.80, 65.48, 49.93, 30.53. HRMS (ESI) *m/z* calcd for C$_{10}$H$_{12}$O$_8$$^+$: 261.0610 [M + H]$^+$; found: 261.0606.

### 3.3. Synthesis of 6-O-(R)-3-hydroxybutyril Ascorbic Acid 5: Direct Transesterification of the Methyl (R)-3-hydroxybutyrate 4 with Ascorbic Acid 2

The ascorbic acid **2** (0.1 g, 0.57 mmol) was added to *t*-butanol (8 mL) and the mixture was shaken (orbital shaker) at 60 °C until **2** was completely dissolved. Then, methyl (*R*)-3-hydroxybutyrate **4** (0.135 g, 1.14 mmol), 5 Å molecular sieves (0.8 g), and N 435 (80 mg) were added to the solution and the mixture was shaken at 60 °C. Samples (0.5 mL) were periodically withdrawn, evaporated under vacuum, and analyzed by $^1$H-NMR in order to determine the conversion and verify the absence of the byproduct **6**. After 3.5 h, the mixture was passed through a 7-mesh filter in order to remove the molecular sieves and then through filter paper to recover the N 435 (both filters were washed with chloroform). After solvent evaporation under reduced pressure (20 mmHg), the excess of 4 was recovered by vacuum distillation (2 mm Hg). The residue was submitted to preparative HPLC purification (see following paragraph), which afforded the pure compound **5** (0.85 g, 3.25 mmol) in 57% yield. $[\alpha]_D^{20}$ = +26.9. (c 0.4, CH$_3$OH). $^1$H NMR (400 MHz, dmso) δ 4.65 (d, *J* = 1.2 Hz, 1H), 3.99 (d, *J* = 6.4 Hz, 1H), 2.35 (d, *J* = 1.8 Hz, 1H), 1.08 (d, *J* = 6.2 Hz, 4H).3.99 (d, *J* = 6.4 Hz, 2H), 4.02−3.90 (m, 2H), 2.38 (s, 1H), 2.35 (d, *J* = 1.8 Hz, 1H), 2.35 (d, *J* = 1.8 Hz, 1H), 1.08 (d, *J* = 6.2 Hz, 3H); 13C NMR (101 MHz, dmso) δ 171.40, 170.83, 152.75, 118.60, 75.40, 65.88, 64.80, 63.82, 44.33, 23.83. HRMS (ESI) *m/z* calcd for C$_{10}$H$_{14}$O$_8$$^+$: 263.0767 [M + H]$^+$; found: 263.07531.

### 3.4. Preparative HPLC Purification of Product 5

The crude reaction mixture (feed) was dissolved in $H_2O$ + 0.05% formic acid (FA) with a concentration of 1 g/L. An amount of 42 mL of this solution was loaded into the column and it was eluted with a mixture of water/acetonitrile + 0.05% FA as mobile phase. The flow rate was set at 1 mL/min and the percentage of organic modifier (acetonitrile) was varied as follows: (i) isocratic at 1% for 6 CV; (ii) gradient elution from 1% to 20% in 11 CV; (iii) stripping and washing at 90% for 6 CV before being decreased to the initial value. During gradient elution, the eluate was periodically collected (1 fraction/minute) and analyzed through offline HPLC to evaluate purity. Offline analytics was performed on an Agilent 1100 HPLC (Agilent, Santa Clara, CA, USA) equipped with a column thermostat, an autosampler, and diode array detector. The temperature was set at 25 °C and the detection wavelength was 254 nm. The analytical column employed was a Zorbax SB-C18 150 × 4.6 mm (L × I.D.) packed with 5 μm C18 particles with a 100 Å pore size. The flow rate was set at 1 mL/min, the injection volume was 5 μL, and the mobile phase was the same mixture employed for preparative chromatography. In this case, the percentage of acetonitrile was varied as follows: (i) isocratic at 1% for 4 min; (ii) gradient elution from 1% to 20% in 30 min; (iii) washing at 90% before being decreased to the initial value.

### 3.5. Synthesis of Methyl (R)-3-O-trimethylsilyl-3-hydroxybutyrate 7b

To a solution of (*R*)-3-hydroxybutyrate **4** (0.2 g, 1.69 mmol) and trimethylsilyl chloride (0.37 g, 3.38 mmol) in anhydrous dichloromethane (3 mL), triethylamine (0.47 mL, 3.38 mmol) was added and the reaction mixture was stirred overnight at room temperature. After dilution with dichloromethane (10 mL) and saturated $NH_4Cl$ (10 mL), the organic phase was separated and the aqueous phase was extracted twice with dichloromethane (10 mL). The combined organic phases were washed with saturated $NaHCO_3$, dried over anhydrous $Na_2SO_4$, and evaporated to afford the product **7b** (55% yield), which was employed without further purification. $^1H$ NMR (400 MHz, $CDCl_3$) δ 4.20−4.14 (m, 1H), 3.65 (s, 3H), 2.48 (dd, *J* = 16.0, 6.1 Hz, 1H), 2.41 (dd, *J* = 16.0, 4.7 Hz, 1H), 1.20 (d, *J* = 6.3 Hz, 3H), 0.4 (s, 9H).

### 3.6. Synthesis of 6-O-(R)-3-hydroxybutyril Ascorbic Acid 5: Transesterification of Methyl (R)-3-O-methoxymethyl-3-hydroxybutyrate 7c and Subsequent Deprotection of the Intermediate 8c

Ascorbic acid **2** (100 mg, 0.57 mmol) was added to t-butanol (8 mL) and the mixture was shaken (orbital shaker) at 60 °C until **2** was completely dissolved. Then, 5 Å molecular sieves (0.8 g), compound **7c** (462 mg, 2.85 mmol), and N 435 (80 mg) were added. The resulting mixture was shaken at 60 °C for 24 h. After that, molecular sieves and N 435 were removed as described above and the mixture was evaporated to afford the crude product **8c** slightly contaminated. $^1H$ NMR (400 MHz, $CD_3OD$) δ $^1H$ NMR δ 4.74 (d, *J* = 1.8 Hz, 1H), 4.64 (q, *J* = 7.0 Hz, 2H), 4.30−4.05 (m, 4H), 3.33 (s, 3H), 2.59 (dd, *J* = 15.3, 7.7 Hz, 1H), 2.53 (dd, *J* = 15.3, 5.2 Hz, 1H), 1.23 (d, *J* = 6.2 Hz, 1H). The crude **8c** was dissolved in methanol/$H_2O$ 5:1 (1.2 mL) and Amberlite® IR120 (50 mg) was added. The mixture was stirred at 55 °C and monitored by TLC ($CHCl_3$/MeOH 7:2 with 1% of acetic acid) until the disappearance of the starting reagent. The resin was removed by filtration and the product **5** (133 mg, 0.51 mmol) was obtained after solvent evaporation in 90% overall yield.

### 3.7. Antioxidant Activity Assay

Photochemiluminescence (PCL) measurements were obtained with PHOTOCHEM® equipment, commercialized by Analytik Jena AG (Jena, Germany). The manufacturer also provided for all reagent kits to be used. The better PCL-assay response was obtained by using the ACL method, which is based on Trolox as the standard reference antioxidant. The PCL calibration curve was prepared in the range of 0.125–2.5 nmol Trolox and the reaction time was set at 360 s. DPPH (2,2-Diphenyl-1-picrylhydrazyl) for the TEAC assay was purchased by Merck. The DPPH calibration curve was between 0.05 and 1 mM Trolox

to obtain a radical inhibition up to 93%. The TEAC assay protocol was applied as reported in previous works [34,35].

## 4. Conclusions

In this work, two unprecedented reported molecular hybrids were synthetized and fully characterized. In spite of the multifunctional nature of the reagents, the use of the commercial lipase B from *Candida antarctica* N435 allowed for obtaining the desired products through simple synthetic procedures. In the new hybrid products, a ketone body moiety, represented by the acetoacetate or the (*R*)-3-hydroxybutyrate, was linked to L-ascorbic acid (2) through a stable but metabolizable ester bond. As both the acyl and the alcoholic moieties of these new ester products have proven activities in the control of oxidative stress and inflammation associated with several chronic neurological and cardiovascular disorders, we think that the results reported herein could stimulate further pharmacological studies. For this reason, a preliminary study on the antioxidant activity of the new products was also conducted.

**Supplementary Materials:** The following supporting information can be downloaded at: https://www.mdpi.com/article/10.3390/catal13040691/s1, Figure S1. $^1$H- and $^{13}$C-NMR spectra of compound **3**; Figure S2. $^1$H- and $^{13}$C-NMR spectra of compound **5**; Figure S3. $^1$H- and $^{13}$C-NMR spectra of compound **8c**; Figure S4. ESI-MS spectra of the reaction mixtures of the esterification of **2** performed with 16 equivalents of **4** (reaction stopped after 3 hours); Figure S5. (a) $^1$H NMR (selected data) of the reaction mixture performed with equimolar amounts of **2** and **4** after 4 hours. (b) $^1$H NMR (selected data) of the reaction mixture performed with 2 equivalents of **4** after 3.5 hours. (c) $^1$H NMR (selected data) of the reaction mixture performed with 3 equivalents of **4** after 3 hours; Figure S6. $^1$H-NMR analyses (selected data) for the time course study of the esterification of **2** performed with 2 equivalents of **4**; Figure S7. HPLC chromatogram of the esterification of 2 performed with 2 equivalents of **4**.

**Author Contributions:** Conceptualization, P.P.G., V.V. and S.A.; methodology, L.A.L., M.C., N.M. and A.B.; formal analysis M.C., N.M. and A.B.; investigation, F.P., S.A. and V.V.; data curation, L.N.D.; writing—original draft preparation, P.P.G.; writing—review and editing, L.N.D.; funding acquisition, P.P.G. and V.V. All authors have read and agreed to the published version of the manuscript.

**Funding:** This research was funded by the National Recovery and Resilience Plan (NRRP), Mission 04 Component 2 Investment 1.5—NextGenerationEU, Call for tender n. 3277 dated 30 December 2021, Award Number: 0001052 dated 23 June 2022, and by the European Union—PON Ricerca e Innovazione 2014–2020 ai sensi dell'art. 24, comma 3, lett. a), della Legge 30 dicembre 2010, n. 240 e s.m.i. e del D.M. 10 agosto 2021 n. 1062.

**Data Availability Statement:** No new data were created.

**Acknowledgments:** We gratefully thank Paolo Formaglio for the NMR experiments and Ercolina Bianchini for the laboratory activity support.

**Conflicts of Interest:** The authors declare no conflict of interest.

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
