# Peer review of "Enzymatic Synthesis of Ascorbic Acid-Ketone Body Hybrids"

_catalysts, doi:10.3390/catal13040691_

Round 1

Reviewer 1 Report (Previous Reviewer 1)

The authors further improved the manuscript based on mine and the other reviewer’s suggestions. However, I still found several issues in the content that the authors should tackle.

1. Line100-101. The related references should be cited.

2. Figure 1 and Figure 2, How many biological repeats are performed by authors. None of error bars are illustrated in the figures. This is important as the data must be convincing.

Author Response

We thanks the Referee for the additional comments. As requested, the new reference 26 has been added in order to support as reported in lines 100-101. New figures 1 and 2 where error bars have been addet to the charts have been produced. The figure's footnotes  explain that the values indicated in the charts come from the mean of three independent experiments.

Reviewer 2 Report (Previous Reviewer 2)

Thank you for addressing all issues raised. 

Author Response

We gratefully thank the Referee for the precious suggestions and for the positive judgment on the revised version of the manuscript.

Reviewer 3 Report (Previous Reviewer 3)

The manuscript has improved over the first submission. 

Author Response

We gratefully thank the Referee for the precious suggestions and for the positive judgment on the revised version of the manuscript.

This manuscript is a resubmission of an earlier submission. The following is a list of the peer review reports and author responses from that submission.

Round 1

Reviewer 1 Report

The manuscript employed lipase B to synthesize ascorbic acid-ketone bodies hybrids.Although I appreciate the general premise of the paper, the case in my view is not sufficiently well presented for further publishing. The major problem is that the experiment in the manuscript is insufficient. For example, the author just mentioned the final conversion yield. The substrate consumption and product titer during the reaction process was not provided. As the data in the manuscript is not convincing, it appears that publication in any form would be premature at this time.

Reviewer 2 Report

The manuscript submitted by Venturi et al. reports the regioselective transesterification of b-keto and b-hydroxyl esters with ascorbic acid by lipase catalysis. All products have been analyzed thoroughly. The products are suggested to have a combined pharmaceutical potential. First tests on the antioxidant activities indicate a moderate antioxidant capacity of the molecular hybrids.

The development of the reaction conditions for substrate 1 are straightforward. As substrate 4 leads to more side reactions I would think that, here, a real reaction optimization would be required.

1)    In line 165 the optimization of reagent ratio and reaction time is mentioned. I think a conversion versus time graph for the reaction between 4 and 2 would be very helpful to find a real optimum in reaction time. It would be even more useful to look in addition at the time course of the side product concentration. Similarly, no optimum can be derived from two ratio variations (16 eq., 2 eq.). Please provide additional data to support that finding.

2)    Oligomerization can often be suppressed by reducing the temperature. Did you try to work at lower reaction conditions or is the temperature of 60° required for the solubility?

3)    Did you perform reference experiments without enzyme under identical reaction conditions? Did you observe any conversion?

4)    I cannot see that the results of the antioxidant assay are particularly promising. A MW- weighted average of 4 and 2 would still give a higher value than the value found for product 5. Please clarify.

Reviewer 3 Report

This manuscript discussed the synthesis of two molecular hybrids of 6-O-ascorbyl esters of acetoacetic and (R)-3-hydroxybutyric acids by using N435 in the transesterification of ketone body methyl esters with ascorbic acid. The two novel ascorbic acid ketone bodies hybrids may be interesting bioactive. I think this is some original in the field of preparation of ascorbyl ketone body hybrids.

The experiments are well designed, the concept is well proved. The tables and figures are standardized.

The downside is that using only one lipase N435, it is best to discuss whether a better enzyme is available or whether it is superior to chemical synthesis methods.

Minor comments

S5 is not clear.

Line 333, 3.7 begins with no spaces.